# Physical-Layer Security with Irregular Reconfigurable Intelligent Surfaces for 6G Networks [note 2]

**DOI:** 10.3390/s23041881

**Published:** 2023-02-07

**Authors:** Emmanuel Obeng Frimpong, Bong-Hwan Oh, Taehoon Kim, Inkyu Bang

**Affiliations:** 1Department of Intelligence Media Engineering, Hanbat National University, Daejeon 34158, Republic of Korea; 2School of Internet of Things, Xi’an Jiaotong-Liverpool University, No. 111, Taicang Avenue, Taicang, Suzhou 215488, China; 3Department of Computer Engineering, Hanbat National University, Daejeon 34158, Republic of Korea

**Keywords:** physical-layer security, 6G, reconfigurable intelligent surfaces, optimization, secrecy rate

## Abstract

The goal of 6G is to make far-reaching changes in communication systems with stricter demands, such as high throughput, extremely low latency, stronger security, and ubiquitous connectivity. Several promising techniques, such as reconfigurable intelligent surfaces (RISs), have been introduced to achieve these goals. An RIS is a 2D low-cost array of reflecting elements that can adjust the electromagnetic properties of an incident signal. In this paper, we guarantee secrecy by using an irregular RIS (IRIS). The main idea of an IRIS is to irregularly activate reflecting elements for a given number of RIS elements. In this work, we consider a communication scenario in which, with the aid of an IRIS, a multi-antenna base station establishes a secure link with a legitimate single-antenna user in the presence of a single-antenna eavesdropper. To this end, we formulate a topology-and-precoding optimization problem to maximize the secrecy rate. We then propose a Tabu search-based algorithm to jointly optimize the RIS topology and the precoding design. Finally, we present simulation results to validate the proposed algorithm, which highlights the performance gain of the IRIS in improving secure transmissions compared to an RIS. Our results show that exploiting an IRIS can allow additional spatial diversity to be achieved, resulting in secrecy performance improvement and overcoming the limitations of conventional RIS-assisted systems (e.g., a large number of active elements).

## 1. Introduction

The goal of 6G is to revolutionize wireless systems with more enhanced requirements, such as very high data rates, very low latency, always-on broadband global network coverage, and intelligence [1]. To achieve these goals, several advanced wireless technologies, such as massive multiple-input and multiple-output (MIMO) transmission, cooperative communications, and cognitive radio, have been proposed. However, these techniques focus on processing signals to adapt to the wireless environment and are, in some ways, affected by the random and negative effects caused by the propagation environment. Recently, a reconfigurable intelligent surface (RIS) was introduced, and it has gained wide attention due to its ability to control the wireless environment. An RIS is a low-cost uniform array of passive reflecting elements where each element can adjust the amplitude and/or phase of any incident signal to make the propagation medium controllable. This characteristic can be exploited to constructively (or destructively) add different signals to enhance (or weaken) the overall signal received [2,3].

In effect, an RIS can be used to enhance the data rate and security, increase coverage, and improve the signal-to-noise ratio (SNR). There are existing RIS-related technologies, such as traditional reflecting surfaces [4], amplify-and-forward (AF) relays [5], and backscatter communication [6,7]. Compared to these techniques, an RIS has the following benefits. Firstly, the traditional reflecting surfaces have fixed phase shifts that cannot be changed, while the RIS is able to continuously change its phase shift. Secondly, while AF relays require a certain amount of power for signal propagation, an RIS passively reflects incident signals without any transmission power consumption. This makes RISs green and energy efficient. Thirdly, even though both RISs and backscatter communication implement the idea of passive communication, an RIS can be equipped with a large number of reflecting elements, while backscatter communication is usually equipped with few antennas due to cost and complexity limitations [8]. These significant advantages have gained the attention of researchers. RISs have been applied in several communication scenarios, such as massive device-to-device communications, secure communications in wireless networks, and wireless information and power transfer, and they have achieved interesting results.

Additionally, RISs have been implemented to tackle physical-layer security (PLS) issues in wireless systems, which make up one of the main research areas in wireless communications. Wireless signals pose broadcasting characteristics during transmission. Hence, security is one of the major concerns in future 6G networks, and with the advent of quantum computing, eavesdroppers have been poised to pose huge threats to wireless systems [9]. Several works integrating RISs and PLS have been introduced [8,9,10,11,12,13,14,15,16,17,18,19,20,21,22,23,24]. However, these studies considered an RIS-assisted system in which all reflecting/refracting elements were activated and employed in transmission (a so-called ‘regular RIS’). With a regular RIS, the system capacity can mainly be improved by increasing the total number of RIS elements [10]. However, this leads to very high channel estimation and feedback overhead, as well as complex beam design. Further, most existing works assumed that the power consumption of an RIS element is negligible. However, with a large number of RIS elements, RIS power consumption could not be negligible. In this regard, the number of RIS elements cannot be too large in real-world systems. Accordingly, in [25,26,27,28], an irregular RIS (IRIS), which significantly improved the system capacity with the limited number of active RIS elements, was introduced. An IRIS guarantees relatively low channel feedback and estimation overhead and a lower power consumption. To the best of our knowledge, the application of the notion of the IRIS to secure communication has been less investigated and studied compared to security problems in regular RIS-assisted systems, as discussed. Accordingly, we tackled a physical-layer security problem in an IRIS environment. Our main contributions are summarized as follows:We investigate a secure communication scenario in which, with the help of an IRIS, a multi-antenna base station wants to transmit data to a legitimate single-antenna receiver in the presence of a single-antenna eavesdropper. Our aim is to guarantee secrecy by designing an effective precoder at the BS and the reflecting beamformer via the phase shifters at the IRIS.We formulate a secrecy rate maximization problem to jointly design the precoder at the BS and the reflection coefficients at the IRIS in order to maximize the secrecy rate for given constraints.We then propose a Tabu-search-based algorithm that can design the topology and precoders for the IRIS to heuristically achieve the maximum performance by efficiently avoiding the local maximum issue for a given formulated non-convex optimization problem.Finally, through simulations, we verify that exploiting an IRIS with a limited number of reflecting elements can significantly improve the secrecy rate compared to that achieved with a regular RIS.

The rest of this paper is outlined as follows. Section 2 summarizes several related studies and compares them with our work. In Section 3, we describe our system model, including the eavesdropping scenario. We formulate the optimization problem to maximize the secrecy rate between the transmitter and receiver while considering the IRIS in Section 4. We discuss our proposed algorithm for solving the optimization problem in Section 5. The simulation results are provided in Section 6, and conclusions are finally drawn in Section 7.

## 2. Related Works

RISs have become some of the essential candidates for 6G wireless communication systems due to their ability to control random wireless environments by manipulating their reflecting coefficients and their potential for various 6G applications. In this regard, extensive research has been conducted in which several RIS-aided secure communication scenarios have been considered. In [10], Yang et al. assessed the secrecy performance by using the secrecy outage probability against eavesdropping threats in an RIS-aided single-input and single-output (SISO) environment. They showed that the secrecy rate increased with the number of reflecting elements on the RIS. Chu et al. [11] considered an RIS-assisted multiple-input and single-output (MISO) wiretap environment, designed a power-efficient secure transmission scheme, and showed that the transmitter could use low power to meet secrecy requirements. In [12], Dong et al. considered RIS-aided multi-input, multi-output, and multi-antenna eavesdropper (MIMOME) systems. The authors proposed an alternating-based algorithm to maximize the secrecy rate and showed that secrecy could be achieved in a non-line-of-sight environment when perfect channel state information (CSI) of the receiver and eavesdropper was available at the base station (BS) and the RIS, respectively. Sun et al. investigated an RIS-assisted and unmanned aerial vehicle (UAV)-secured network in which a single UAV BS transmitted data to a receiver in the presence of an eavesdropper [13]. The authors proposed algorithms that maximized the secrecy rate by carefully designing the positioning of the UAV BS and the beamforming of the RIS.

The authors of [8,14] considered a scenario in which multiple legitimate users were attacked by multiple eavesdroppers. Specifically, in [14], a novel deep-reinforcement-learning-based algorithm was designed to jointly optimize the beamforming at the BS and the RIS while considering different quality of service (QoS) requirements and time-varying channel conditions. Chen et al. [8] investigated a downlink MISO broadcast system in which the BS transmitted an independent data stream to multiple legitimate receivers against multiple eavesdroppers. In particular, the authors considered a special case in which the receivers and the eavesdroppers were in the same direction as the transmitter, with a high correlation between the channels of the receivers and eavesdroppers. They also considered both continuous and discrete reflecting coefficients for the RIS’s elements. Then, they formulated an optimization problem to maximize the minimum secrecy rate. To this end, they proposed an algorithm based on a path-following algorithm to solve the non-convex problem. The authors of [15] proposed a novel artificial noise (AN)-based jamming protocol via an RIS, which showed that AN could enhance the secrecy rate in RIS-aided networks. Dong et al. [16] studied a more robust AN-aided transmission scheme that guaranteed secrecy performance without any knowledge of the eavesdropping CSI. To increase the flexibility of RIS deployment, the concept of the refracting RIS was introduced, where all incident signals could pass through the RIS’s surface, which could effectively reconstruct the channels for blocked users and significantly improve the system performance. In [17], Lin et al. implemented a refracting RIS to improve the quality of service (QoS) in a blocked hybrid satellite–terrestrial relay network (HSTRN). The main goal was to minimize the total transmission power of the satellite and base station while satisfying the rate requirements of multiple users. Furthermore, Niu et al. [18] focused on improving the secrecy rate among users by sending signals through an active refracting RIS-based transmitter and a passive reflecting RIS in the presence of multiple eavesdroppers. In this work, the authors proposed an optimization framework that considered jointly designed the power allocation, the transmission beamforming of the refractive RIS, and the phase shifts of the reflective RIS in order to maximize the weighted-sum secrecy rate. The authors of [19] investigated a secure multibeam satellite system in which a satellite user in each beam was surrounded by an eavesdropper attempting to intercept confidential data. They proposed an alternating optimization scheme that maximized the secrecy energy efficiency under a transmission power constraint.

Xu et al. [20] introduced a simultaneously transmitting and reflecting reconfigurable intelligent surface (STAR-RIS), a flexible RIS design that was able to reflect and transmit incident signals. Unlike a conventional RIS, which only reflects and requires users to be on the same side as the transmitter, the STAR-RIS had the ability to achieve $360^{\circ }$ coverage. The authors showed that the STAR-RIS achieved a higher diversity order on both sides of the surfaces compared to that of a traditional RIS. Accordingly, the STAR-RIS was investigated further in many studies, including PLS-related studies [21,22,23]. In [21], Niu et al. considered a multiuser MISO STAR-RIS-aided secure communication in which energy-splitting, time-splitting, and mode-selection transmission protocols were investigated. In each instance, the authors maximized the weighted-sum secrecy rate by jointly optimizing the beamforming and the transmitting and reflecting coefficients of the STAR-RIS. In [22], the authors focused on a STAR-RIS-assisted uplink-secure non-orthogonal multiple-access (NOMA) communication, where they considered both full and statistical eavesdropping CSI. Han et al. [23] introduced AN into secure NOMA communication in STAR-RIS networks. The authors proposed an alternating optimization (AO)-based algorithm that derived the optimal AN model and the RIS parameters for maximizing the secrecy rate.

The concept of using a limited number of RIS elements (IRIS) was introduced by Su et al. in [25]. In this work, an IRIS was implemented to improve the system capacity with a limited number of RIS elements. The main idea was to irregularly configure a given number of RIS elements on an enlarged surface. To this end, they formulated a topology and beamforming design problem to maximize the system capacity. The authors extended their work in [26] by considering several communication scenarios involving time-varying channels, but they still focused on maximizing the system capacity. Chen reformulated the joint IRIS topology and beamforming design problem as an equivalent reflecting beamforming design problem in which the RIS element selection mechanism was embedded into the coefficient of each RIS element [27]. Then, the author proposed a probability-learning algorithm to solve the formulated problem. The author of [28] investigated the capacity maximization problem in an IRIS-aided multiuser MISO downlink system by jointly optimizing the phases of all RIS elements, the transmission beamforming at the base station, and the RIS topology. The author further considered activating only a small portion of elements in order to reduce the complexity and power consumption. In summary, IRIS-assisted systems have been investigated less than RIS-assisted systems. In addition, to the best of our knowledge, physical-layer security issues have not been investigated in detail while considering IRIS-assisted systems, although wireless security becomes more important than ever in 6G networks.

## 3. System Model

Figure 1 describes an existing regular RIS network and the considered irregular RIS network. For both, we consider three nodes in the network: a base station, a legitimate user (Bob), and an eavesdropper (Eve). The existing regular RIS-assisted system, in which a BS with *M* antennas and a regular RIS with $N_{s}$ elements transmit data to a receiver (i.e., Bob) in the presence of an eavesdropper, is shown in Figure 1a. The regular RIS has all active reflecting elements arranged with a constant spacing on a regular surface. We consider an IRIS-assisted secure communication system. Figure 1b shows an example of an irregular RIS with *N* active elements selected out of $N_{s}$ elements $\left(N\leq N_{s}\right)$. Let Z=diag(z) denote the $N_{s}\times N_{s}$ matrix representing the topology of the RIS, where $z={\left[z_{1},z_{2},\cdots ,z_{N_{s}}\right]}^{T}$. We define $z_{n}\in \left\{1,0\right\}$ as an indicator to represent the state of the *n*-th reflecting element in the IRIS, where ‘1’ indicates the active state (i.e., the corresponding RIS element is activated) and ‘0’ is used otherwise.

The reflection coefficient matrix for $N_{s}$ grid points of the IRIS is given by
(1)$$\begin{array}{c} \Theta =\mathrm{diag}\left(\left[\beta_{1}e^{j\theta_{1}},\beta_{2}e^{j\theta_{2}},\cdots ,\beta_{N_{s}}e^{j\theta_{N_{s}}}\right]\right), \end{array}$$
where $\beta_{n}$ and $\theta_{n}$ represent the reflection amplitude and phase shift of the *n*-th RIS element in the IRIS.

For practical hardware implementation, we assume a constant reflection amplitude and finite discrete phase-shift constraint at the RIS [24]. Specifically, $\forall n\in \{1,2,\cdots ,N_{s}\}$, $\beta_{n}=1$, and $\theta_{n}$ takes discrete values from the quantized phase-shift set given by
(2)$$\begin{array}{c} F=\left\{0,\frac{2\pi }{2^{b}},\cdots ,\frac{2\pi }{2^{b}}\left(2^{b}-1\right)\right\}, \end{array}$$
where *b* is the number of bits used to quantize the finite discrete phase shifts.

The received signal at the legitimate receiver (i.e., Bob) is expressed as
(3)$$\begin{array}{c} y_{b}=h_{b}^{H}Z\Theta \mathrm{Gw}x+n_{b}, \end{array}$$
where *x* denotes the transmitted signal at the BS with the power constraint ${∥x∥}^{2}\leq 1$, $n_{b}$ indicates additive white Gaussian noise (AWGN) with zero mean and variance $\sigma_{b}^{2}$, $G\in C^{N_{s}\times M}$ is the channel coefficient matrix between the BS and RIS, $h_{b}\in C^{N_{s}\times 1}$ denotes the RIS–Bob channel coefficient vector, and $w\in C^{M\times 1}$ indicates the precoding vector.

Similarly, the received signal at the eavesdropper (i.e., Eve) is expressed as
(4)$$\begin{array}{c} y_{e}=h_{e}^{H}Z\Theta \mathrm{Gw}x+n_{e}, \end{array}$$
where $n_{e}$ indicates additive white Gaussian noise (AWGN) with zero mean and variance $\sigma_{e}^{2}$ and $h_{e}\in C^{N_{s}\times 1}$.

Note that the precoding vector $w\in C^{M\times 1}$ in (Equation 3) and (Equation 4), the reflection coefficient matrix Θ, and the topology of the IRIS Z (i.e., activation and inactivation pattern of the RIS elements) are our main design factors for maximizing the secrecy performance for the given system environment.

Thus, the secrecy rate of the IRIS-assisted secure communication is a function of w, Θ, and Z, and it is given by
(5)$$\begin{array}{c} R_{s}\left(w,\Theta ,Z\right)={\left[\log_{2}\left(1+\frac{{∥h_{b}^{H}Z\Theta \mathrm{Gw}∥}^{2}}{\sigma_{b}^{2}}\right)-\log_{2}\left(1+\frac{{∥h_{e}^{H}Z\Theta \mathrm{Gw}∥}^{2}}{\sigma_{e}^{2}}\right)\right]}^{+} \end{array}$$
where ${\left[\cdot \right]}^{+}=\max\left\{0,\cdot \right\}$.

## 4. Problem Formulation for IRIS-Assisted Secure Communication

In this section, we introduce the problem formulation for IRIS-assisted secure communications. Note that the secrecy performance of the proposed IRIS-assisted system in (Equation 5) is a function of the IRIS topology and precoding. Thus, the topology and precoding should be carefully designed. In the following, we formulate the secrecy rate maximization. Let w denote the fully digital precoder at the BS. The transmission power at the BS is
(6)$$\begin{array}{c} P_{T}={∥w∥}^{2}. \end{array}$$

$P_{T}$ in (Equation 6) should be lower than or equal to the maximum allowable transmission power *P*. The secrecy rate maximization problem can be formulated as
(7)$$\begin{array}{cc} (P1): & \max_{Z,w,\Theta }\;R_{s}\left(w,\Theta ,Z\right)\\ = & \max_{Z,w,\Theta }\;{\left[\log_{2}\left(1+\frac{{∥h_{b}^{H}Z\Theta \mathrm{Gw}∥}^{2}}{\sigma_{b}^{2}}\right)-\log_{2}\left(1+\frac{{∥h_{e}^{H}Z\Theta \mathrm{Gw}∥}^{2}}{\sigma_{e}^{2}}\right)\right]}^{+} \end{array}$$
(8)$$\begin{array}{c} \mathrm{subject}\mathrm{to}\;(C1):\;P_{T}\leq P, \end{array}$$
(9)$$\begin{array}{c} \;(C2):\;\theta_{n}\in F,\forall n=1,2,\cdots ,N_{s}, \end{array}$$
(10)$$\begin{array}{c} \;(C3):\;z_{n}\in \left\{1,0\right\},\forall n=1,2,\cdots ,N_{s}, \end{array}$$
(11)$$\begin{array}{c} \;(C4):\;1^{T}z=N, \end{array}$$
where (**C1**) denotes the transmission power constraint, (**C2**) denotes discrete phase-shift constraint, and (**C3**) and (**C4**) denote the constraint for the topology of the irregular RIS activation, where there are *N* ones (i.e., activation of the element) and $N_{s}-N$ zeros (i.e., deactivation of the element) in the topology matrix Z.

Note that the objective function in (12), the phase-shift constraint (**C2**), and the IRIS topology constraints (**C3**) and (**C4**) are non-convex. Hence, directly solving (**P1**) is a non-trivial task. To address this issue, we reformulate (**P1**) for a given IRIS topology by separating the precoding design and the IRIS topology design. Then, for a particular topology $Z_{p}$, the original problem (**P1**) reduces to
(12)$$\begin{array}{cc} (P2): & \max_{w,\Theta }\;R_{s}\left(w,\Theta ,Z_{p}\right)\\ = & \max_{w,\Theta }\;{\left[\log_{2}\left(1+\frac{{∥h_{b}^{H}Z_{p}\Theta \mathrm{Gw}∥}^{2}}{\sigma_{b}^{2}}\right)-\log_{2}\left(1+\frac{{∥h_{e}^{H}Z_{p}\Theta \mathrm{Gw}∥}^{2}}{\sigma_{e}^{2}}\right)\right]}^{+} \end{array}$$
(13)$$\begin{array}{c} \mathrm{subject}\mathrm{to}\;({C1}^{\prime }):\;P_{T}\leq P, \end{array}$$
(14)$$\begin{array}{c} \;({C2}^{\prime }):\;\theta_{n}\in F,\forall n=1,2,\cdots ,N_{s}, \end{array}$$
(15)$$\begin{array}{c} \;({C3}^{\prime }):\;Z=Z_{p}, \end{array}$$
where (**C1**’) and (**C2**’) denote the transmission power constraint and discrete phase-shift constraint, respectively, and (**C3**’) denotes the given IRIS topology.

Note that irregular RIS-assisted systems can be considered as regular RIS-assisted systems by assuming that $N=N_{s}$ and $Z=I_{N}$. In other words, the activated elements in the irregular RIS are all assumed to be elements in the regular RIS. Fortunately, we can efficiently solve (**P2**) by using our proposed algorithm and can suboptimally solve (**P1**) for given IRIS topology candidates, which will be discussed in detail in the next section.

## 5. Proposed Algorithm for IRIS-Assisted Secure Communication

In this section, we introduce our proposed algorithm for solving (**P1**). In order to efficiently solve (P1), we propose an adaptive algorithm based on the Tabu search algorithm [29]. Tabu search is a metaheuristic searching algorithm that can efficiently avoid local minima (or maxima) for given optimization problems by exploiting diversification rules to drive the search into new regions of the searching space. Figure 2 briefly illustrates the main idea of our proposed algorithm, and the details are summarized in Algorithm 1.

In Algorithm 1, for the given $N_{s}$ RIS elements, the RIS topology is irregularly initialized with a random selection of *N* active elements and $N_{s}-N$ inactive elements at line 2 (i.e., $Z_{i}$). Note that we consider $I_{N}$ random topology initializations, and each initialization dynamically changes the active and inactive elements together. After the IRIS topology initialization at line 2, we consider $I_{T}$ minor topology changes for the given IRIS topology. For this, we randomly swap *v* ones for *v* zeros in the given $Z_{i}$ at line 5. At line 5, $Z_{\left(i,t\right)}$ denotes an instance of the IRIS topology at a given initialization and a given swap. We try these random swaps for $I_{T}$ times for the given $Z_{i}$. Unlike an initialization, $Z_{\left(i,t\right)}$ has only minor changes compared with $Z_{i}$.

Note that (**P1**) is reduced to (**P2**) for a given $Z_{\left(i,t\right)}$. We solve (**P2**) by using *K* samples of $\Theta^{k}$ at line 7. The effective channels for all candidates are calculated at line 8. At lines 9 and 10, the maximum ratio of transmission precoding is assumed, and the secrecy rates are calculated for all *K* samples. We then select the candidate with the maximum secrecy rate ($R_{\mathrm{temp}}$) and compare $R_{\mathrm{temp}}$ with $R_{\max}$, which is updated at every iteration in the second for-loop after the initialization. After the second for-loop, we update the topology set *T* to avoid duplication. Finally, we select the best topology in the topology list with the maximum secrecy rate and use this one for the IRIS operation.
**Algorithm 1** Adaptive IRIS Operation for Secrecy Enhancement**Require:**$\;G,h_{b},h_{e}$, number of iterations $I_{N}$ and $I_{T}$, number of samples *K*, swap value *v*,  phase-shift set F, topology set T**Ensure:** RIS Topology $Z_{\max}$1:**for**$\;i=1:I_{N}$**do**2:    Initialize the RIS topology Z by randomly selecting *N* ones and $N_{s}-N$ zeros: $Z_{i}$3:    Initialize $R_{\max}=0$4:    **for** $t=1:I_{T}$ **do**5:        Randomly swap *v* ones for *v* zeros among the diagonal elements of $Z_{i}$: $Z_{\left(i,t\right)}$6:        /* We are ready to solve (**P2**) for the given $Z_{\left(i,t\right)}$ */7:        Randomly generate *K* samples ${\left\{\Theta^{k}\right\}}_{k=1}^{K}$8:        Calculate the effective channel for Bob and Eve9:        Calculate the precoder’s coefficients ${\left\{w^{k}\right\}}_{k=1}^{K}$10:      Calculate the secrecy rate ${\left\{R_{s}\left(w^{k},\Theta^{k},Z_{\left(i,t\right)}\right)\right\}}_{k=1}^{K}$11:      $R_{\mathrm{temp}}=\max_{\forall k}{\left\{R_{s}\left(w^{k},\Theta^{k},Z_{\left(i,t\right)}\right)\right\}}_{k=1}^{K}$12:      **if** $R_{\mathrm{temp}}>R_{\max}$ **then**13:           $R_{\max}=R_{\mathrm{temp}}$14:           $k^{★}=\arg\max_{\forall k}{\left\{R_{s}\left(w^{k},\Theta^{k},Z_{\left(i,t\right)}\right)\right\}}_{k=1}^{K}$15:      **end if**16:    **end for**17:    Set $Z_{i}=Z_{\left(i,k^{★}\right)}$, $w=w^{k^{★}}$, $\Theta =\Theta^{k^{★}}$18:    Add $Z_{i}$ with $R_{s}\left(w,\Theta ,Z_{i}\right)$ in T/* $\left(Z_{i},R_{s}\left(w,\Theta ,Z_{i}\right)\right)\in T$ */19:**end for**20:$i^{★}=\arg\max_{\forall i}\left\{R_{s}\left(w,\Theta ,Z_{i}\right)\right\}$ for $\forall Z_{i}\in T$21:Return $Z_{\max}=Z_{i^{★}}$

### Computational Complexity Analysis

It is important to discuss the computational complexity when we consider non-convex optimizations such as (**P1**). By using the proposed Algorithm 1, we can find the best IRIS topology for the given parameters.

We solve (**P1**) by considering sub-problem (**P2**), and solving (**P2**) requires $K\times 2^{bN}$ computations. We repeat this to solve (**P2**) for $I_{N}\times I_{T}$ IRIS topology candidates. Thus, it requires an approximate total of $I_{N}I_{T}K\times 2^{bN}$ computations to solve (**P1**). Practically, it is important to analyze the complexity based on the number of RIS elements $N_{s}$, since the size of the RIS is a critical issue for system performance and deployment. Our proposed algorithm can provide constant computational complexity O(1) in the best case when we set $I_{N}$, $I_{T}$, *K*, and *N* to be independent of $N_{s}$, where O(·) represents big O notation for computational complexity analysis [30]. However, it is worth noting that the computational complexity can be $O(2^{N_{s}})$ in the worst case if we consider all of the possible topology candidates (i.e., $I_{N}\times I_{T}=2^{N_{s}}$).

**Remark 1.** 
*Note that our proposed algorithm is designed based on Tabu search, a metaheuristic searching algorithm that is commonly used to solve non-convex optimizations [29]. In the proposed algorithm, we exploit the diversification to explore a new searching space and, thus, heuristically approach the suboptimum by avoiding local maxima. Our algorithm searches $I_{N}$ possible RIS topologies and $I_{T}$ swap cases. Thus, for a given topology and a swap case, we have $K\times 2^{bN}$ candidates in each iteration. Therefore, the total number of candidates in our proposed algorithm is $I_{N}I_{T}K\times 2^{bN}$, and it can be set by manually adjusting $I_{N}$ and $I_{T}$. The proposed algorithm reaches an exponential computational complexity in the worst case, but usually, we can achieve better performance compared with that of a regular RIS, even for constant values of $I_{N}$ and $I_{T}$ and a small value for N. Thus, the proposed algorithm can even be applicable to the hundreds of reflecting elements in an RIS, since we only activate a limited number of reflecting elements in the RIS. Further, the proposed algorithm requires less computation and power consumption during channel estimation and overall operation, respectively, compared to an RIS.*


## 6. Numerical Results

In this section, we numerically evaluate the performance of the proposed algorithm on irregular RIS-assisted communication systems. We first introduce our simulation setup and discuss simulation results in detail.

### 6.1. Simulation Setup

For the simulations, we considered an independent Rayleigh fading channel model to account for the small-scale fading of all channel coefficients, where each channel coefficient followed a Gaussian distribution with zero mean and unit variance. For the large-scale fading, we considered the effective path loss while considering antenna gain and distances together. The large-scale fading coefficients of BS–IRIS–Bob and BS–IRIS–Eve are expressed, respectively, as
(16)$$\begin{array}{cc} L_{b} & =\Gamma_{b}d_{\mathrm{BR}}^{-\alpha_{\mathrm{BR}}}d_{\mathrm{RU}}^{-\alpha_{\mathrm{RU}}}, \end{array}$$
(17)$$\begin{array}{cc} L_{e} & =\Gamma_{e}d_{\mathrm{BR}}^{-\alpha_{\mathrm{BR}}}d_{\mathrm{RE}}^{-\alpha_{\mathrm{RE}}} \end{array}$$
where $d_{BR}$, $d_{RU}$, and $d_{RE}$ are the distance between BS and IRIS, the IRIS and Bob, and RIS and the eavesdropper respectively. $\Gamma_{b}$ and $\Gamma_{e}$ represent the effects of channel fading and antenna gain on Bob’s and Eve’s channels respectively. Additionally, we assume that CSI is available at the BS. The path loss exponent is denoted by $\alpha_{BR},\alpha_{RU}$, and $\alpha_{RE}$. We set the parameters as follows: $\Gamma_{b}=\Gamma_{e}=-60$ dB, $\alpha_{BR}=2$, and $\alpha_{RU}=\alpha_{RE}=3$. We set the distances between the BS and Bob and the BS and Eve as $d_{E}=145$ m and $d_{B}=150$ m, respectively. The distance between the RIS and Eve was set to $d_{RE}=5$ m. Then, the distance between the BS and RIS was $d_{\mathrm{BR}}=\sqrt{\left(d_{E}^{2}+d_{RE}^{2}\right)}$. We set the noise power to $\sigma_{b}^{2}=\sigma_{e}^{2}=-80$ dBm. We considered the quantized bit number b=1. For the proposed algorithm, we considered $I_{N}=10$, $I_{T}=40$, K=200, and v=3. During the simulations, we considered a regular RIS as a baseline scheme for comparison with our proposed algorithm for the irregular RIS (i.e., IRIS).

### 6.2. Secrecy Performance

In this subsection, we take a look at the secrecy rate performance of the IRIS-assisted secure communication system in comparison with that of a regular RIS.

Figure 3 shows the secrecy rate versus transmission power when the BS was equipped with multiple antennas. All elements (i.e., $N_{s}=40$) were activated for the regular RIS, but only half of the elements (i.e., N=20) were activated for the irregular RIS (IRIS). In other words, we implemented the IRIS with $N_{s}=40$ elements, out of which we randomly activated N=20 elements. We considered two cases in which the base station was equipped with multiple antennas: M=4 and M=8 (in Figure 3a) or M=16 and M=32 (in Figure 3b). The secrecy rate increased for both the regular RIS and irregular RIS as the transmission power increased. We observed in both cases that the IRIS outperformed the regular RIS in the high-transmission-power regime (≥30 dBm), even though we only considered half of the activated reflecting elements in the IRIS. This was because exploiting the IRIS allowed us to achieve additional spatial diversity in terms of reflecting elements; thus, it contributed to boosting the secrecy performance of the IRIS. In addition, these performance gains between the regular RIS and IRIS increased when the transmission power increased: 17% (30 dBm) →69% (40 dBm) for M=4, 5% (30 dBm) →42% (40 dBm) for M=8, 2% (30 dBm) →34% (40 dBm) for M=16, and 10% (30 dBm) →37% (40 dBm) for M=32. As the number of antennas increased, the performance gain between the regular RIS and IRIS decreased, since we only considered transmission beamforming [31]. Note that the IRIS could effectively improve the secrecy rate with a limited number of reflecting elements when we considered high transmission power and a lower number of antennas.

Figure 4 shows the impact of the number of active elements on the secrecy rate. We set the transmission power to $P_{T}=30$ dBm. We set the swap value to v=2 and reduced it to 1 after half of the number of iterations for fine-tuning. The regular RIS with all $N_{s}=20$ elements was used as a benchmark. In Figure 4a, we compare the secrecy rate performance of different regular RIS scenarios with that of the IRIS. We simulated the IRIS with M=4 antennas. We also considered several regular RIS-based cases in which the BS was equipped with M=4 and M=8 antennas. We set the total number of RIS elements to $N_{s}=20$ and randomly activated $N<N_{s}$ reflecting elements. Firstly, we observed that exploiting the IRIS can be a solution for the improvement of secrecy performance by using a limited number of reflecting elements. In addition, we observed that the IRIS with M=4 antennas underperformed when 2∼4 elements were randomly activated, but performed better when we activated more than six elements. Further, we observed that the secrecy performance of the IRIS with M=4 was almost close to that of the RIS with M=8 antennas. Specifically, if active elements in the IRIS with M=4 were carefully selected (e.g., 18 out 20 elements are activated in Figure 4a), the secrecy performance could be maximized. Similarly, in Figure 4b, we simulated the IRIS with an increased number of antennas: M=4 to M=32. We also reconsidered the baseline regular-RIS-based cases in which the BS was equipped with M=32 and M=64 antennas. In this simulation, we set the total number of RIS elements to $N_{s}=20$ and randomly activated $N<N_{s}$ reflecting elements. We can observe similar trends in Figure 4a,b. The secrecy rate of the IRIS with M=32 antennas underperformed for 2∼8 active elements of the IRIS, but performed better when we activated more than 10 elements of the IRIS.

In Figure 4a,b, it can be observed that the secrecy rate was improved when the reflecting elements were selected randomly. As we increased the number of active elements in the IRIS, the secrecy rate increased, slowed down, and eventually dropped, approaching that of the regular RIS. Thus, in practice, an IRIS should be carefully designed to guarantee the desired secrecy rate. Even though the RIS with more antennas (e.g., M=64) performed better compared with the IRIS with fewer antennas (e.g., M=32), we could achieve almost the same secrecy rate as that of the RIS equipped with more antennas by using the IRIS with only a lower number of antennas. Note that, for given Ns, the minimum required number of active elements needed to outperform the secrecy performance of the RIS increased when the number of antennas at the BS increased. This was because the performance gain between the regular RIS and IRIS decreased as the number of antennas increased, as we discussed in Figure 3.

### 6.3. Discussion

In this subsection, we briefly mention the channel estimation and scheduling issues of our work from the perspective of the medium access control (MAC) layer. In order to effectively exploit the advantages of irregular reconfigurable intelligent surfaces, it is important to accurately estimate the channels to and from the RIS. However, this is subject to challenge, since it requires advanced signal processing techniques, such as indirect estimation of channel coefficients due to the passive elements of the RIS. Recently, several studies investigated signal processing and channel estimation techniques for RISs [32,33]. In addition, from the perspective of the MAC layer, the frame structure should be carefully designed while relying on the duplexing mode, pilots, and multiuser scheduling [34]. Accordingly, designing advanced MAC layer protocols for jointly considering channel estimation, user scheduling, and the operation of irregular reconfigurable intelligent surfaces could be one of the possible future research directions of our work.

## 7. Conclusions

In this paper, we investigated guaranteeing secure communications by using an IRIS. The secrecy rate of traditional RIS systems is restricted by the number of elements in the RIS. Specifically, in order to increase the secrecy rate, the number of elements should be increased. However, this might not be practical. To tackle this issue, we analyzed an irregular RIS (i.e., IRIS) with a limited number of elements to enhance the secrecy performance. To this end, we formulated a secrecy rate maximization problem and proposed an optimization framework to solve it. Through simulations, we validated that an IRIS that used only a limited number of elements could sufficiently enhance the secrecy rate past that of a regular RIS. We observed that by using a lower number of reflecting elements that were randomly distributed on an RIS, we could achieve an extra degree of freedom, which could enhance the secrecy performance. For example, the secrecy rate of an irregular RIS increased by 42% at a 40 dBm transmission power with eight transmission antennas. This was achievable with relatively low channel estimation and feedback overhead if we used the proposed algorithm. We also noticed that the number of active elements should be carefully selected during system design. Several open problems exist and are left for future work. For example, increasing the quantization bits, considering continuous phase shifts, and designing MAC layer protocols for IRIS-assisted secure communication systems can be some of the possible future research directions.

## Figures and Tables

**Figure 1 sensors-23-01881-f001:**
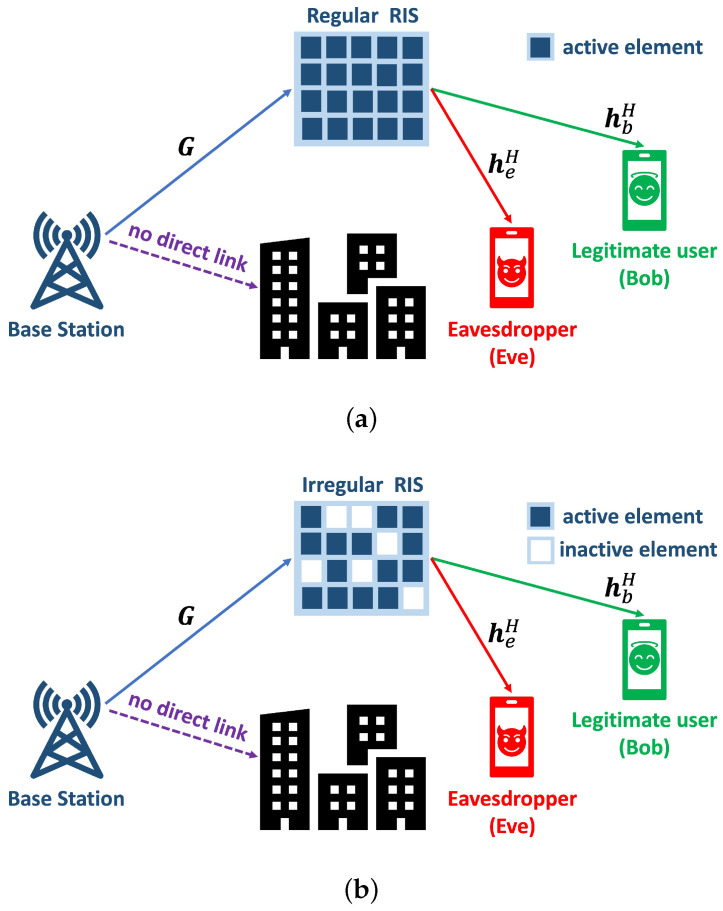
System model: RIS-assisted secure communication. (**a**) Regular RIS. (**b**) Irregular RIS.

**Figure 2 sensors-23-01881-f002:**
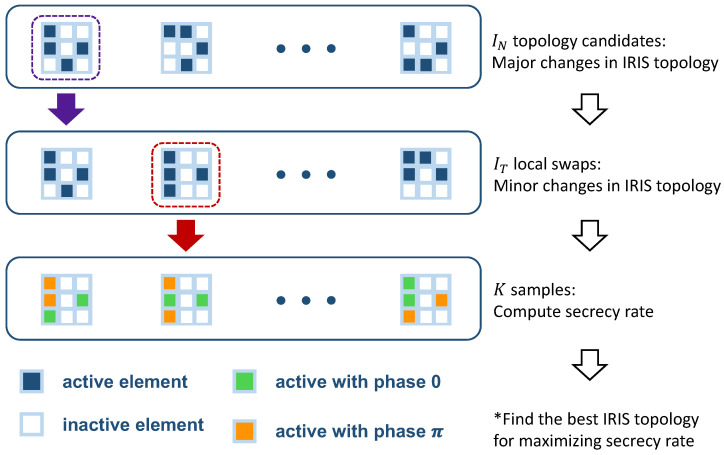
Illustration of the proposed algorithm.

**Figure 3 sensors-23-01881-f003:**
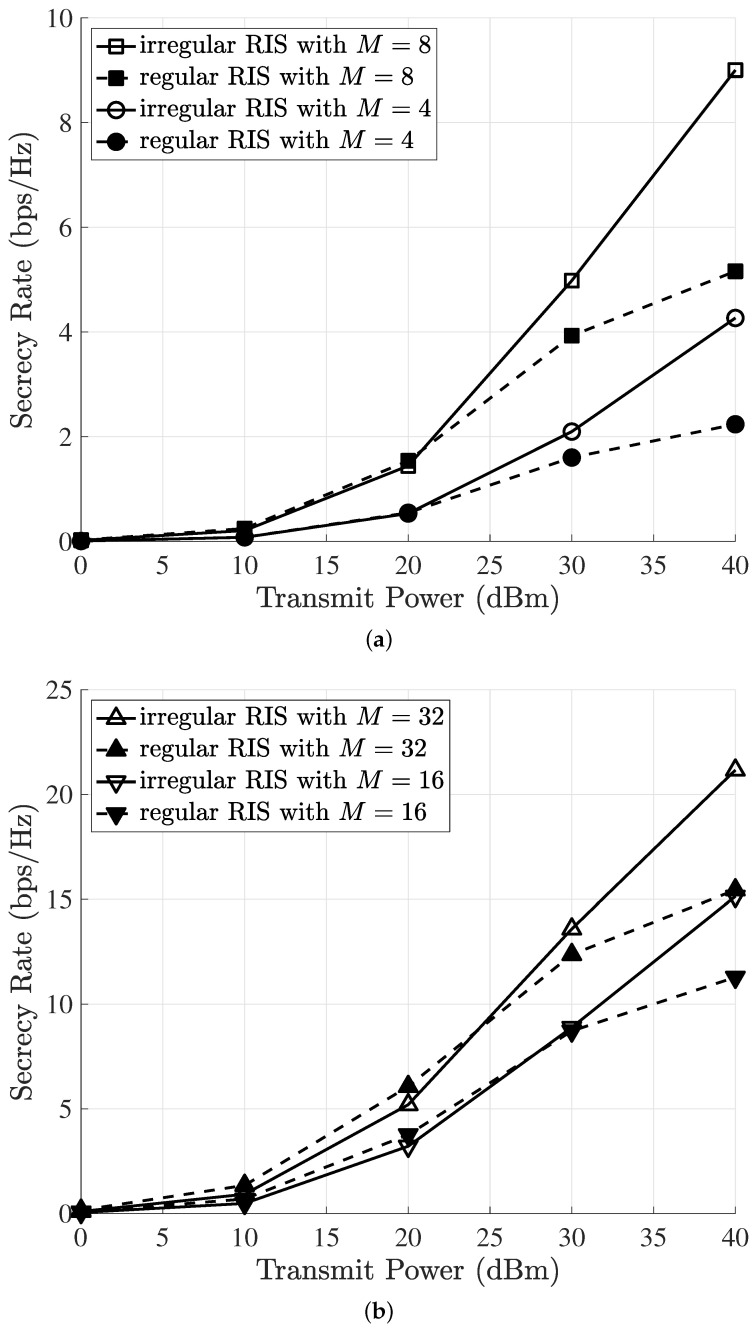
Secrecy rate versus transmission power when the BS was equipped with multiple antennas: M=4, M=8, M=16, and M=32. (**a**) Secrecy rate for M=4, M=8, N=20, and $N_{s}=40$. (**b**) Secrecy rate for M=16, M=32, N=20, and $N_{s}=40$.

**Figure 4 sensors-23-01881-f004:**
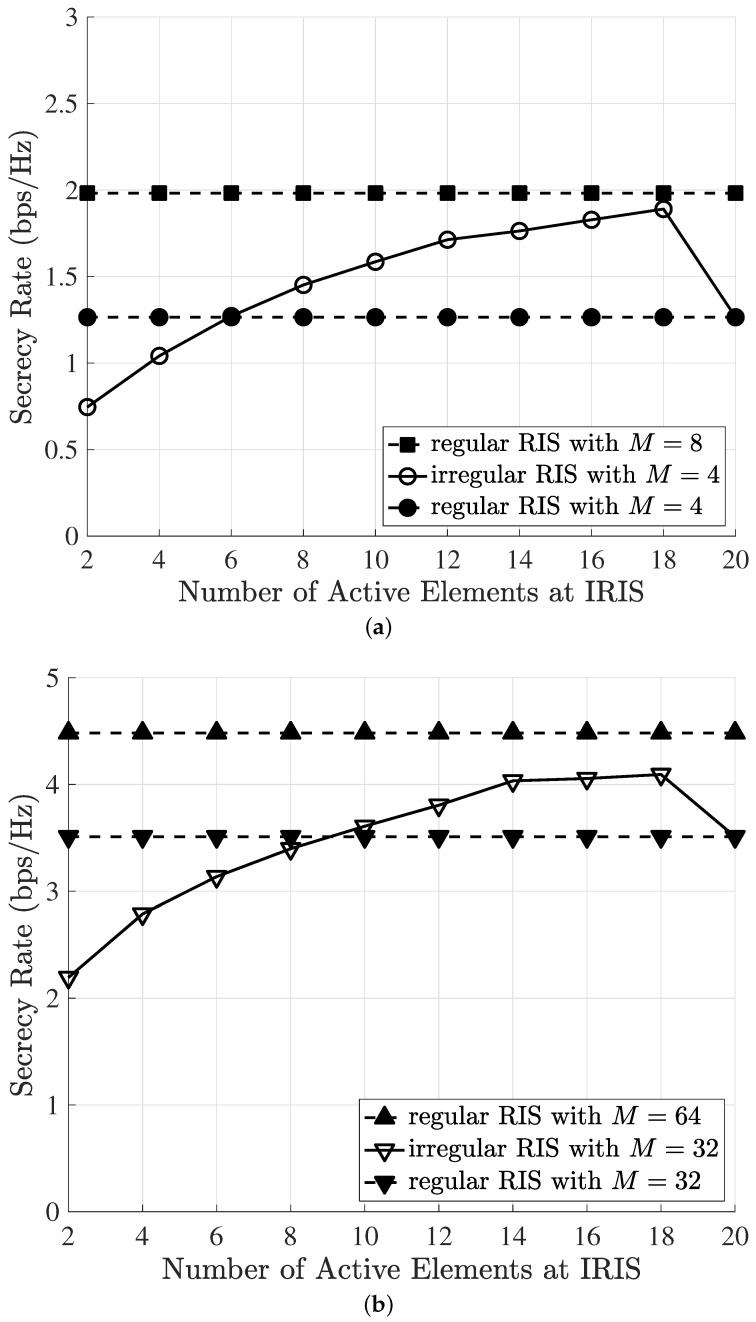
Secrecy rate versus the number of active reflecting elements at $N_{s}=20$ and $P_{T}=30$ dBm. (**a**) Secrecy rate of the irregular RIS with M=4 and the regular RIS for reference with M=4,8. (**b**) Secrecy rate of the irregular RIS with M=32 and the regular RIS for reference with M=32,64.

## Data Availability

Not applicable.

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
