# Peer review of "Physical-Layer Security with Irregular Reconfigurable Intelligent Surfaces for 6G Networksâ€"

_sensors, 2023, doi:10.3390/s23041881_

Round 1
Reviewer 1 Report
Please see the attached file.

Author Response
Dear Editor and Reviewer
We genuinely appreciate the valuable comments from the reviewer and editor. We have done our best to reflect all the concerns addressed in the review. We believe that the manuscript has been further improved. In addition, the concerns mentioned by the reviewer have been addressed in our revised manuscript (the revised parts are highlighted in blue color in the revised manuscript) and answered in this letter. Please refer to the revised manuscript and this reply letter.
Please note that we have separately made a reply letter for each reviewer since the submission system requires uploading individual responses to each reviewer.
**Please see the attachment.

Reviewer 2 Report
This paper needs a major revision,
- Please indicate the insights obtained by your work
- Please simulate with large M, e.g., 32,64
- Please survey more papers on physical layer security of RIS. The following paper should by cited: Artificial Noise Aided Secure NOMA Communications in STAR-RIS Networks
Author Response

(The authors gave the same response as above.)

Reviewer 3 Report
First, in terms of content, it is estimated at the level of a conference paper, and both in terms of the number of references and attention to other layers - beyond just the physical layer - the text should be strengthened. Secondly, a similar article by the first three of the four present authors can be seen at the following address:
So it is necessary for the authors to explain that the current work is really so different from this article that they considered it suitable for submission in the journal?!
Author Response

(The authors gave the same response as above.)

Round 2
Reviewer 1 Report
The authors have addressed my concerns, no further comments.
Author Response
Thank you for your comments.
Reviewer 2 Report
thank you for your revision
Author Response
Thank you for your comments.
Reviewer 3 Report
In part 2, which you mentioned about learning algorithms, the following paper should be cited:
https://www.nature.com/articles/s41371-018-0052-3
Author Response
We have tried to understand Reviewer 3's comment but we think the suggested link by Reviewer 3 is incorrect. The research area of the suggested paper is quite different from ours.Title of the suggested paper: "Less primary fistula failure in hypertensive patients"
Research Area: Journal of Human Hypertension
Title of our paper: "Physical-Layer Security with Irregular Reconfigurable Intelligent Surfaces for 6G Networks"
Research Area: Wireless Communication